Decoupled recovery of ecological communities after reclamation

Sylvain Zachary A. zach.sylvain@gmail.com
Branson David H.
Rand Tatyana A.
West Natalie M.
Espeland Erin K.
Pest Management Research Unit, USDA—Agricultural Research Service , Sidney , MT , United States of America
Stuble Katharine
Electronic publication date: 2019 Jun 21
Publication date: 2019
Volume: 7
Electronic Location ID: e7038
Received 2018 Sep 13; Accepted 2019 Apr 29
Copyright year: 2019
License: This is an open access article, free of all copyright, made available under the Creative Commons Public Domain Dedication. This work may be freely reproduced, distributed, transmitted, modified, built upon, or otherwise used by anyone for any lawful purpose.
License URL: https://creativecommons.org/publicdomain/zero/1.0/

Keywords: Alternative stable states, Biotic interactions, Hysteresis, Nematodes, Plants, Soil abiotic factors

Funding: USDA-ARS project 5436-22000-017-00D 3032-21220-002-00-D The authors received funding from in-house appropriated USDA-ARS project Nos. 5436-22000-017-00D and 3032-21220-002-00-D. The funders had no role in study design, data collection and analysis, decision to publish, or preparation of the manuscript.

==============================
Grassland restoration is largely focused on creating plant communities that match reference conditions. However, these communities reflect only a subset of the biodiversity of grassland systems. We conducted a multi-trophic study to assess ecosystem recovery following energy development for oil and gas extraction in northern US Great Plains rangelands. We compared soil factors, plant species composition and cover, and nematode trophic structuring between reclaimed oil and gas well sites (“reclaims”) that comprise a chronosequence of two—33 years since reclamation and adjacent, undeveloped rangeland at distances of 50 m and 150 m from reclaim edges. Soils and plant communities in reclaims did not match those on undeveloped rangeland even after 33 years. Reclaimed soils had higher salt concentrations and pH than undeveloped soils. Reclaims had lower overall plant cover, a greater proportion of exotic and ruderal plant cover and lower native plant species richness than undeveloped rangeland. However, nematode communities appear to have recovered following reclamation. Although total and omni-carnivorous nematode abundances differed between reclaimed well sites and undeveloped rangeland, community composition and structure did not. These findings suggest that current reclamation practices recover the functional composition of nematode communities, but not soil conditions or plant communities. Our results show that plant communities have failed to recover through reclamation: high soil salinity may create a persistent impediment to native plant growth and ecosystem recovery.

Introduction

Restoration of human-degraded landscapes aims to recover ecological communities and their functions, and is critical for sustainable ecosystem management (Hobbs & Harris, 2001). Conventional restoration focuses on reestablishing the most common members of the plant community (Suding, 2011), setting the stage for natural colonization of the area by other taxa (i.e., the “field of dreams” paradigm; Hilderbrand, Watts & Randle, 2005). Restoration is intended to eventually reflect reference (e.g., undisturbed and natural) ecosystems, while reclamation may only restore a subset of reference conditions. Most studies of reclamation focus on individual ecosystem components; a multi-trophic approach is integral to understanding the interactions that influence or limit ecosystem recovery from disturbance.

Disturbed and reclaimed environments (hereafter referred to as “reclaims”) present several impediments to the recovery of plant communities. Post-disturbance plant communities are often either depauperate (as in old-field systems) or lacking altogether (e.g., recent agricultural fields or surface mines), and native plant species must be seeded onto the site (i.e., assisted colonization, restoration, or reclamation). However, planted seed mixes have lower species diversity compared to the adjacent landscape (e.g., Simmers & Galatowitsch, 2010). Additionally, undesirable (i.e., invasive, exotic, or ruderal) plants are often present in disturbed areas, either seeded intentionally during reclamation, incidentally by transport on machinery, or through natural colonization.

Numerous ecosystem components interact with plant communities to drive the ecosystem functions present in reference sites, such as soils and their communities (see Wardle et al., 2004; Sylvain & Wall, 2011). Restoration and reclamation practices and studies often overlook soil characteristics and soil organisms (Snyder & Hendrix, 2008). Soil organisms are intimately tied to soil abiotic conditions and plant community composition and structure, with the latter being especially important due to the dependence of soil food webs on plant-derived resources (i.e., litter inputs, root exudates and root material). Although plants are often seeded during reclamation, soil biota are not commonly introduced (Lawrence et al., 2013) and must, therefore, colonize from adjacent landscapes. Recovery of these communities is slow (Murphy & Foster, 2014; Viall et al., 2014; Wodika & Baer, 2015) and may depend upon soil microstructure and the plant community (Baer, Heneghan & Eviner, 2012). Among soil organisms, nematodes are a group that may be particularly useful in monitoring studies (such as responses to disturbance and recovery) owing to their numerous interactions with plants and the soil habitat (Neher, 2001; Sylvain & Wall, 2011). They are also easily extracted and sorted into functional/trophic groups (Yeates et al., 1993) and their trophic positioning, life history (especially lifespan relative to shorter-lived soil microbes) and sensitivity to environmental conditions facilitates their use in characterizing ecosystem conditions along disturbance gradients (Bongers & Ferris, 1999; Neher, 2001). The broad span of trophic levels and comparatively longer lifespans of nematodes make them uniquely suited to use as indicators of environmental press dynamics such as those occurring during environmental recovery (as opposed to more transient pulse dynamics that strongly influence microbial communities). Nematodes involved in decomposition pathways (see Moore & De Ruiter, 2012) can respond to and even out the microbial responses to environmental pulses (such as ephemeral nutrient additions) and herbivorous nematodes can provide an indication of the prevalence of root material on which they feed (Neher, 2001; Sylvain & Wall, 2011). Together, these factors serve to reflect the state of nutrient cycling and revegetation dynamics and can be used to indicate how successfully restoration is progressing.

Soil conditions are often degraded in reclamations, either during the initial disturbance event or as a consequence of management. For example, topsoil stockpiling and redistribution damages soil structure and depletes soil communities (Schuman, 2002; Menta, 2012). Destruction of soil aggregates, increased infiltration rates, reductions in soil biota, and plant root growth into the stockpiled soil eventually leads to losses of carbon and nutrients, reducing accumulated topsoil productivity (Menta, 2012), although site to site variability is extremely high (Emam, Espeland & Rinella, 2014). Compaction during reclamation may also inhibit plant establishment and growth (Bassett, Simcock & Mitchell, 2005). In addition to soil structural changes, several aspects of soils disturbed by stockpiling may contribute to the presence of environmental filters that impede recovery of plant and soil communities on restored areas. These can include comparatively low amounts of soil organic matter (Viall et al., 2014), leading to reduced nutrient availability for plant production. Subsoils are stockpiled separately from topsoils and in either stockpile type high salinity may result from integration of concentrated salt layers typical in this region (Espeland & Perkins, 2013).

Energy development for oil and gas is a common type of land-use change across arid and semi-arid grasslands of central North America. Over the last few decades, increased development of these areas has resulted in an average of 50,000 new oil and gas wells per year (Allred et al., 2015), transforming the landscape. Oil and gas well sites are supported by road and pipeline infrastructure, creating a matrix of degraded habitat within rangelands, disrupting landscape connectivity and providing increased opportunity for introductions of undesirable species to both native rangeland and nearby cropland (Viall et al., 2014; Preston, 2015). As a consequence, Allred et al. (2015) estimated total productivity losses due to these activities at 10 Tg dry plant biomass. Reclamation could mitigate detrimental impacts (e.g., increased erosion, reduced water infiltration, species invasions) and return these landscapes to the productivity of reference sites. Despite the importance of reclamation for the sustainable management of these landscapes, there has been no systematic assessment of the mechanisms by which current oilfield reclamation recovers the composition and functional structure of ecological communities and their interactions.

We present one of the first studies to employ a multi-trophic assessment of reclamation outcomes. We examined reclamation in semi-arid grasslands following energy development in the Bakken oil and gas fields of western North Dakota, USA. We assessed whether reclamation has successfully returned soils or plant and nematode communities to reference conditions, comparing sampling locations on reclaims up to 33 years old with locations on adjacent, intact prairie both close to (50 m) and distant from (150 m) reclaim edges (“undeveloped rangeland”). We hypothesized that (1) reclaimed plant communities would have greater cover of undesirable plant species, more bare ground and decreased species richness than adjacent, intact prairie. We also hypothesized that (2) undesirable plant species would increase on close intact prairie transects with time as they spread from reclaimed well sites. As a consequence of soil stockpiling and redistribution during reclamation activities, we hypothesized that (3) soil organic matter (SOM) would decrease and that soil salinity and pH would increase on reclaimed sites compared to adjacent, undeveloped rangeland sites. We further hypothesized that (4) nematode community recovery on reclaims would be closely linked to patterns in soil conditions and plant community composition and structure across transects. Specifically, we predicted that nematode abundances would decrease as salinity and pH increased (less suitable habitat for nematode persistence); reduced resource availability would reduce populations of root herbivorous nematodes (due to reduced available root biomass as bare ground increased) and bacterivorous and fungivorous nematodes (due to reductions in SOM caused by increased bare ground and low litter cover). Finally, because reclamation does not involve moving nematodes to a site in the way plants are seeded into reclamations, we hypothesized that (5) nematode community recovery would lag that of plant communities due to slower dispersal.

Materials & Methods

Site selection and study design

We conducted field sampling within the northern half of the Little Missouri National Grassland (between 47.404°N–47.743°N and 103.394°W–104.041°W). This region of the northern mixed-grass prairie in western North Dakota is characterized as wheatgrass-needlegrass prairie (Barker & Whitman, 1988) with mean annual precipitation of 387 mm and mean annual temperature of 6.8 °C (Aziz, Champa & VanderBusch, 2006). We selected fourteen reclaimed oil and gas well sites (hereafter “reclaims”) on public land designated either “plugged and abandoned” or “dry” with well and pipeline shapefiles provided by the North Dakota Industrial Commission, Department of Mineral Resources, Oil and Gas Division (NDIC) using ArcGIS (v. 10.2.1, ESRI, Redlands, California, USA). Records on the seeding history of our sites were so poor they were unusable; a variety of reclamation practices were likely employed, contributing to variability in plant communities on reclaims. Sites were selected to have edges a minimum of 250 m from pipelines and access roads and to minimize cover of bare ground, rugged terrain (e.g., draws and steep hillslopes) and creeping juniper (Juniperus horizontalis Moench) in order to compare consistent plant types (e.g., grasses) and topography. Finally, sites that had historically been tilled were excluded to avoid confounding effects of disturbances from sources other than energy development. All map layers were exported from ArcGIS to Google Earth Pro (v. 7.1.4.1529, Google Inc., Mountain View, California, USA) prior to site selection. Our selected reclaims ranged in age since reclamation (when a site was released from bond after being deemed sufficiently revegetated for cattle exclosure fencing to be removed) from two to 33 years (Fig. S1).

In order to assess the effects of energy development and revegetation disturbance on rangelands, 100 m transects were established on reclaims as well as on adjacent undisturbed prairie at two distances,50 m and 150 m, from the reclaim edge. As the undisturbed prairie transects have maintained continuous native prairie cover, these sites are all of a similar age with each other and it is therefore only the reclaim sites that can be assigned to a particular “age” since reclamation. At each of the 14 sites, three parallel transects were placed using Google Earth followed by ground-truthing for final transect location (n = 42 transects). Transect placement was designed to help detect potential movement of undesirable plant species from reclaims into native rangeland or recolonization of reclaims by native rangeland species.

Plant and soil sampling

We sampled plant communities between 25 July and 25 August 2016. A Garmin GPSMap64st was used to identify the start and end points of the 100 m transect. At 15 evenly spaced points along the transect, on alternating sides, a (20 cm × 50 cm) Daubenmire frame was placed at a random distance up to 7.5 m. Within each frame, we measured plant species cover (Anderson, 1986) to the nearest 5% increment (total cover could total >100%). Species were identified and determined to be native or exotic, and were further divided into invasive (exotics only) or ruderal (both natives and exotics), using Johnson & Larson (2007), Larson & Johnson (2007) and Stubbendieck, Hatch & Butterfield (1997). It is important to note that both native and exotic plants can follow a ruderal life history strategy, and that not all exotic species are necessarily invasive. Research showing the abundance of undesirable species in disturbed habitats and eventual population expansion into adjacent sites has often not distinguished between ruderal (e.g., Grime & Mackey, 2002) and invasive species (e.g., Spellerberg, 1998; Trombulak & Frissell, 2000; Simmers & Galatowitsch, 2010) though persistence of these separate plant types has different implications for the health of reclaims. Although colonization and persistence of ruderal species are associated with disturbance (Grime & Mackey, 2002; Espeland & Perkins, 2017), invasive species are able to establish and persist, spread spatially, and have detrimental ecological impacts in both undisturbed and disturbed environments (Dietz & Edwards, 2006; Richardson et al., 2000). Additional cover classes included bare ground and litter. We sorted plants into functional groups including annual and perennial grasses, forbs, shrubs and subshrubs (i.e., dwarf shrubs with persistent woody stems but seasonally-limited herbaceous growth). At each Daubenmire frame location, a 6 cm diameter soil sample was collected to a depth of 10 cm (i.e., the biologically active layer), and these samples were used to create two bulk soil samples per transect (one bulk sample for each half of the 100 m transect) in 1 gallon zip-top bags. Soils were returned to the lab, oven dried at 30 °C, sealed tightly with excess air removed and shipped to Ward Laboratories, Inc. (Kearney, NE, USA) for chemical and physical analysis. Additionally, soil compaction was measured using a soil compaction tester (agraTronix, Streetsboro, OH, USA) at 14 points along each transect at two depths, 7.5 cm and 15 cm, and then averaged across each transect within each depth.

Nematode sampling

We collected soil samples for nematode analyses between 20 September and 28 October 2016. Each transect described above was used to create two contiguous 50 m × 20 m plots (i.e., two half-transect plots) centered along the 100 m transect. Ten soil cores 2.5 cm diameter and 10 cm depth were collected from locations placed haphazardly and aggregated into a single bulk sample, sealed in zip-top bags with excess air removed and transported to the laboratory in coolers. These bulk samples were subsampled for nematode extraction and soil moisture measurement. Nematodes were extracted using the Baermann funnel method (Baermann, 1917) with 50 g of soil over 72 h, and soil moisture was determined gravimetrically using 50 g soil heated to 105 °C for 48 h in an oven and calculating mass loss due to evaporation (Barrett et al., 2008). Live nematodes were counted and assigned to trophic groups according to Yeates et al. (1993) using an Olympus CKX53 inverted microscope at 100–200× and were then preserved in 5% formalin solution. Although extraction using Baermann funnel has been shown to have certain drawbacks in the efficiency with which the method recovers certain nematode groups (generally being biased in favor of extracting greater numbers of smaller and more active individuals; Freckman, Mankau & Ferris, 1975), other methods also show biases in extraction efficiency (see McSorley & Frederick, 2004). For broad ecological studies such as ours where advantages of one method over another are unclear, Baermann funnel extraction is a commonly used approach (e.g., Garcia-Palacios et al., 2017; Sylvain & Mosseler, 2017; Andriuzzi & Wall, 2018).

Analyses

Prior to analyses, nematode counts and soil chemical and physical characteristics were averaged to yield single values per transect. Plant cover was averaged across all frames within each transect. Nematode abundances were standardized to the number of individuals per kilogram dry soil (using gravimetrically determined soil moisture content). Additionally, an aggregate trophic grouping of “omnicarnivores” was calculated by summing omnivore and predator abundances. We performed indicator species analysis using the labdsv package (Roberts, 2016). This analysis identifies species characteristic of predetermined groups of sites (Dufrene & Legendre, 1997), and we used this approach to select a subset of plant species characteristic of sampling locations in our study (i.e., reclaims, and undeveloped rangeland at 50 m and 150 m distances from reclaims) for use in subsequent analyses (species with a significant indicator value ≥0.45). Plant species identified as indicators using this analysis included two from undeveloped rangeland, Carex filifolia (native sedge, 150 m sampling location) and Bouteloua gracilis (native grass, 50 m sampling location), and four from the reclaims, Elymus trachycaulus (native grass), Agropyron cristatum (exotic and invasive grass), Medicago lupulina (exotic and ruderal forb) and Distichlis spicata (native grass).

Analyses were conducted in R version 3.2.4 (R Core Development Team, 2016) and all data were tested to meet assumptions of normality using QQ plots and the Shapiro–Wilk test; where data were found to be non-normal, response variables were transformed using either log (x + 1) or square root transformations. Code used for all analyses is available at GitHub (https://github.com/Ofmitesandmen/Bakken-plants-soils). All analyses on plant data were conducted on the most commonly encountered species (present in >5% of sampling quadrats). We used mixed-model ANOVA to address hypotheses related to plant (H1), soil property (H3) and nematode (H4) responses to energy development and reclamation. Testing was carried out using the lme4 package (Bates et al., 2015), and the Kenward-Roger approximation was used to calculate degrees of freedom for F statistics and associated P-values in the lmerTest package (Kuznetsova, Brockhoff & Christensen, 2016) using type III sums of squares. Our models used sampling location (i.e., transects on reclaims or on undeveloped rangeland at 50 and 150 m from reclaim edges) as a fixed effect and site as a random blocking effect in order to control for landscape variation between sites; response variables included plant species richness (total, native and exotic), plant cover (natives, exotics, invasives, ruderals, total grasses, total forbs and species identified using indicator species analysis), bare ground, soil abiotic factors (salts, pH, CEC, SOM, nitrate, texture and compaction) and nematode abundances (total and individual trophic groups). Post-hoc tests were carried out using the multcomp package (Hothorn, Bretz & Westfall, 2008) and adjusted using the Holm method.

Regression testing was carried out using the base R statistics package. To address whether undesirable plants increased with time since reclamation on undeveloped rangeland (H2) and whether nematode community recovery lagged that of the plant community (H5), we ran an initial set of regression models exploring potential interactions between distance from reclaims (using transect type as an ordinal variable with the reclaim set at 0) and time since reclamation. As undisturbed prairie transects (at 50 m and 150 m from reclaim edges) represent sites that have not undergone conversion from native prairie, these analyses enabled us to determine whether native plants have moved from undisturbed prairie to recolonize reclaims over time as well as whether invasive and exotic plants have moved from reclaims onto undisturbed prairie over time since reclamation (Fig. S1). These regressions used the same response variables as in ANOVA testing. A second set of regression models was run to test hypotheses regarding the relationships among soils, plants, and nematodes (H4). For these models, response variables included all plant and nematode variables tested with ANOVA, and predictors included soluble salt concentrations, SOM, % silt, CEC, and soil pH. For nematodes, predictor variables also included cover of selected plant indicator species, namely crested wheatgrass (Agropyron cristatum [L.] Gaertn.), blue grama (Bouteloua gracilis [Willd. ex Kunth] Lag. ex Griffiths), threadleaf sedge (Carex filifolia Nutt.), bare ground and litter. All final ANOVA and multiple regression models were determined using backward selection and AIC model comparisons to select the most parsimonious model that best fit the data; initial and final models are included in relevant tables.

Nonmetric multi-dimensional scaling (NMS) ordinations using the Bray-Curtis distance measure and PERMANOVA analyses were conducted in the vegan package (Oksanen et al., 2016) to explore dissimilarity among sampling locations in the plant and nematode community data. We also conducted a PERMDISP analysis on the variance around the centroid of each sampling location.

To further explore direct and indirect relationships between plants, soils and nematodes (H4), we constructed a Structural Equation Model (SEM) using the lavaan package (Rosseel, 2012). SEM is a multivariate analytical approach employing path and factor analyses to compare hypothetical models with data (Grace, 2006). Path coefficients are calculated for each set of connected variables in the model, representing the effect of a one standard deviation change in the independent variable on the dependent variable (if all other variables are held constant; Mitchell, 1992). Goodness-of-fit for hypothetical models is compared using χ2 statistics, with non-significant (P > 0.05) values indicating adequate fit between model and data, supporting the null hypothesis represented by the model (Mitchell, 1992). We constructed our initial hypothetical model to represent broad plant community groups (ruderal plants and both native and exotic non-ruderal plants, hereafter simply “natives” and “exotics”), soil variables shown to have strong influence on the biotic community (salts and SOM), the amount of bare ground and litter within transects (common non-plant cover factors), and a nematode community summary variable (created from the first axis of a PCA using nematode community data; 37% variation explained). Our initial model (Fig. S2) represented a saturated interaction web and our final model was constructed iteratively by adapting best-fit models until a model having both a solid mechanistic basis and good fit was found.

Results

Plant community dynamics

We identified 62 native plant species and 11 exotic plant species (common species listed in Table S1). Indicator species analysis identified one species characteristic of 150 m transect type, one characteristic of 50 m transect type, and four characteristic of reclaim transects (including exotic, invasive, and ruderal plant species; Table S2, Fig. 1). ANOVA testing (Table 1) revealed that total plant species richness was lower on reclaims compared to undeveloped rangeland, driven by lower native plant species richness on reclaims (Fig. 2A). Exotic plant species richness was greater on reclaims than on undeveloped rangeland (Fig. 2B). Native plant species cover was also significantly lower on reclaims compared to undeveloped rangeland (Fig. 2C), with exotic plants making up a correspondingly greater proportion of cover on reclaims. Reclaims also had more bare ground (Fig. 2D) and ruderal plant cover (Fig. 2E) and lower cover of grasses (Fig. 2F) compared with undeveloped rangeland. All the ruderals commonly found in our sampling locations were forbs. No significant differences were observed for invasive plant cover across sampling locations. Regression analyses found no effect of time since reclamation on plant community variables except for exotic species richness, where a significant interaction between time since reclamation and distance from reclaims was found (P = 0.0019, R2 = 0.27). This interaction is likely explained by a decrease in exotic species richness with time on the 150 m sampling locations (P = 0.018, R2 = 0.39) and a suggestive (but not significant) trend (P = 0.094, R2 = 0.22) of increasing exotic species richness with time on reclaims. NMS ordination of plant community species composition revealed clear differences between communities associated with reclaims and those associated with undeveloped rangeland (three-dimensional final stress, 0.167; first two dimensions shown, Fig. 3), which PERMANOVA analysis revealed to be significant (P = 0.003 for both native prairie locations compared with reclaims). Similar results were obtained for plant community functional composition (three-dimensional final stress, 0.127, PERMANOVA P = 0.003 for both undeveloped rangeland locations compared with reclaims). PERMDISP analysis revealed significant differences in dispersion about sampling location centroids when comparing plant species composition between undeveloped rangeland locations with reclaims (adjusted P = 0.0004 for 50 m and 0.04 for 150 m undeveloped rangeland locations), but not when comparing the two undeveloped rangeland locations with each other (adjusted P = 0.2). No significant differences in dispersion were found for plant guilds (adjusted P = 0.1 for both undeveloped rangeland locations when compared with reclaims and 1 when comparing undeveloped rangeland locations with each other). Graphic examination of PERMDISP results suggest that while communities are distinct at the species level (the findings of PERMANOVA), reclamation decreases the heterogeneity of plant communities.

Figure 1 Boxplots of selected plant indicator species.

Indicators are characteristic of (A) 150 m sampling locations (Carex filifolia, threadleaf sedge); (B) 50 m sampling locations (Bouteloua gracilis, blue grama); and (C) an invasive plant (Agropyron cristatum, crested wheatgrass) and (D) a native plant (Distichlis spicata, desert saltgrass) from reclaims (“PAD”).

Table 1 ANOVA and regression results for plant community variables.

All ANOVA testing conducted on 2 and 26 degrees of freedom with sampling location (reclaim, 50 and 150 m native prairie transects) as a fixed effect. Significant ANOVA results indicate response variables differ across sampling locations.

Response variable	ANOVA F-statistic	P	Initial regression model	Final regression model	P	R2	
Species richness (total)	7.28	0.003	Salts	–	–	–	
Species richness (native)	13.65	<0.0001	Salts + % Exotic Cover	−% Exotic Cover	0.0001	0.3	
Species richness (exotic)	10.98	0.0003	Salts + % Exotic Cover	Salts + % Exotic Cover	0.0009	0.27	
Native plant cover	21.41	<0.0001	Salts	–	–	–	
Proportion exotic plant cover	6.04	0.007	SOM	–	–	–	
Ruderal plant cover	11.47	0.0003	SOM + Salts + Bare	Bare	0.001	0.24	
Invasive plant cover	2.14	0.14	SOM	SOM	0.01	0.15	
Grass cover	4.37	0.023	Salts + SOM	SOM − Salts	<0.0001	0.45	
Forb cover	1.45	0.25	Salts + SOM + Silt	Silt	0.019	0.13	
Carex filifolia	8.75	0.001	Sodium	−Sodium	0.02	0.14	
Bouteloua gracilis	12.93	0.0001	Calcium + CEC	−CEC	0.04	0.1	
Elymus trachycaulus	8.36	0.002	Salts + SOM + pH	−SOM	0.045	0.1	
Agropyron cristatum	10.56	0.0004	SOM	–	–	–	
Medicago lupulina	9.04	0.001	SOM + Salts	−SOM	0.0008	0.25	
Distichlis spicata	8.48	0.001	Salts + CEC	Salts	0.001	0.23	
Bare ground	5.97	0.007	SOM + Salts + CEC + pH	Salts + pH − SOM	<0.0001	0.54	

Figure 2 Differences in vegetation cover between reclaims (“PAD”) and adjacent, intact rangeland 50 m and 150 m from reclaim edges.

Figure 3 NMS ordination of plant community composition by sampling location.

Open squares represent reclaimed well sites, triangles represent adjacent, intact rangeland 50 m (filled triangles) and 150 m (open triangles) from reclaim edges.

Soil characteristics and nematode community dynamics

Several soil chemical characteristics differed significantly between reclaims and undeveloped rangeland as revealed by ANOVA testing (Table 2). CEC, soluble salts, Ca, Na and pH were all higher and % silt was lower on reclaims compared with undeveloped rangeland (Fig. 4). Compaction at 15 cm (but not 7.5 cm) was significantly greater on reclaims compared with undeveloped rangeland. No differences were found among transects for SOM or soil nitrate. Regression analyses revealed no significant changes to soil abiotic factors with time since reclamation, except for increased compaction at 7.5 cm (P = 0.03, R2 = 0.35) and a suggestive (but not significant) increase in compaction at 15 cm (P = 0.054, R2 = 0.28).

Despite observed differences in soil properties between reclaims and undeveloped rangeland, few differences were observed for nematodes (Table 3; Table S3). Total nematode and omnicarnivorous nematode abundances significantly increased from reclaims to 150 m transect types, with abundances on the 50 m transects intermediate to both (Fig. 5). No other nematode trophic group differed between reclaims and undeveloped rangeland. Despite the differences observed in omnicarnivore and total nematode abundances, neither NMS ordination nor PERMANOVA analysis revealed differences in nematode community structure between reclaims and undeveloped rangeland, and regression testing revealed no interactions between distance from reclaim edge and time since reclamation (although bacterivorous nematodes increased with time on all transect types; P = 0.03, R2 = 0.12).

Interactions between abiotic factors and the biotic community

Regression analyses were used to test hypotheses regarding the relationship among soils, plants and nematodes. There were few strong relationships and in some instances no relationship was found (Table 1). The strongest relationships (R2 > 0.30) involved: bare ground increasing with salt concentrations and soil pH but decreasing with SOM; grass cover increasing with SOM but decreasing with salt concentrations; and native plant richness decreasing with greater proportional cover of exotic plants. Salt concentrations and SOM were most commonly related to abundances of plant species highlighted with indicator species analysis, while forbs were the only plant group to respond to changes in soil texture (decreasing as the proportion of silt declined). There were significant relationships between nematode abundances and a combination of soil and plant community variables, although total variance explained by the models was generally low (15–22%, Table 3). Total abundance of nematodes increased with soil pH but decreased with increasing bare ground, as did abundance of root herbivorous nematodes. Both bacterivorous and fungivorous nematode abundances increased with SOM, while omnicarnivorous nematodes increased with threadleaf sedge, the sole group where abundance was related to a specific plant.

We used SEM analysis to examine direct and indirect interactions between a subset of soil variables shown to strongly influence the biotic community, a summary variable for nematode community, several important plant groups (ruderals, natives and exotics), and other common non-plant ground cover types related to plant cover and disturbance (litter and bare ground). Our initial model (Fig. S2) did not provide a good fit to the data (χ2 = 37.88, df = 7, P < 0.001), and was simplified using an iterative process to find the best fit model (Fig. 6; χ2 = 3.88, df = 7, P = 0.794). Among plant community groups, exotic plants were positively affected by SOM and increased amounts of bare ground, and were negatively affected by native plants. Ruderal plants were positively affected by increased salt concentrations and cover of bare ground. The nematode community did not respond to changes to the plant community groups, but was negatively affected by increased amounts of bare ground. Strengths of the direct effects are summarized in Table S4. Our model also revealed that SOM and bare ground indirectly influence native plants, mediated through direct interactions with exotic plants. The magnitude of the indirect effect of SOM on native plants (−0.61 = 0.69∗ − 0.89) was roughly equivalent to the magnitude of the corresponding direct effect, while the magnitude of the indirect effect of bare ground on native plants (−0.35 = 0.39∗ − 0.89) was twice as strong as the corresponding direct effect. The final model explained a large amount of variation in community variables: 76% for native plants, 44% for ruderal plants, 42% for nematodes, and 34% for exotic plants.

Discussion

This study is one of the first to integrate plant and soil communities with soil characteristics in an assessment of reclamation following energy development for oil and gas extraction. Our data demonstrate that legacies of energy development are still clear in rangelands of western North Dakota, even after 30 years. Soil conditions on reclaims differed from those on adjacent, undeveloped rangeland primarily due to higher pH and salt concentrations and a lower proportion of silt. Plant communities on reclaims had lower total cover (i.e., more bare ground) and greater cover of undesirable plants. There was limited movement of native plants onto reclaims or of invasive or ruderal plants into surrounding rangelands. These results suggest that reclamation has not succeeded in recovering rangeland plant communities or soil conditions. The reduced productivity and greater bare ground cover in reclaims leads to reduced forage availability for livestock and wildlife, as well as increased potential for erosion that can then spread to adjacent lands. In contrast to soils and plants, the functional structure of nematode communities appears to have largely recovered to the reference levels demonstrated by surrounding rangeland.

Table 2 ANOVA results for soil characteristics.

All ANOVA testing conducted on 2 and 26 degrees of freedom with sampling location (reclaim, 50 and 150 m native prairie transects) as a fixed effect. Significant ANOVA results indicate response variables differ across sampling locations and are presented in bold text.

Soil factor	F-statistic	P	150 m	50 m	Reclaim	
			Mean ± SE	Max.	Min.	Mean ± SE	Max.	Min.	Mean ± SE	Max.	Min.	
pH	7.88	0.002	7.5 ± 0.13b	8.1	6.7	7.6 ± 0.10b	8	6.8	7.9 ± 0.04a	8.3	7.7	
Organic matter (%)	2.01	0.15	5.19 ± 0.44	7.25	2.25	5.24 ± 0.34	7.7	2.75	4.13 ± 0.56	10.3	1.45	
CEC (meq/100 g)	13.4	<0.0001	23.11 ± 1.11b	28.4	13.1	24.29 ± 1.33b	29.75	13.45	28.21 ± 1.03a	34.35	17.45	
Ca (ppm)	8.39	0.0015	3465 ± 231b	4,485	1,741	3727 ± 261b	4811	1,721	4343 ± 143a	4940	2,967	
Na (ppm)	5.04	0.014	85.5 ± 37.7b	539	8.5	90.5 ± 46.2b	674	9.5	244 ± 72.5a	755	13.5	
Soluble salts (dS/m)	5.16	0.013	0.35 ± 0.06b	1	0.16	0.39 ± 0.12b	1.9	0.14	0.78 ± 0.19a	2.1	0.18	
Nitrate (ppm)	0.91	0.42	3.76 ± 1.88	27.7	0.6	2.72 ± 0.96	11.95	0.5	3.39 ± 1.05	15.1	0.7	
Sand (%)	0.35	0.71	24.89 ± 4.03	69	9	25.32 ± 3.7	67.5	13	27.07 ± 4.44	71.5	12	
Silt (%)	10.57	0.004	43.11 ± 2.6a	54	16	41.75 ± 2.31a	54	17.5	35.43 ± 2.48b	47	13.5	
Compact. 7.5 cm (MPa)	0.98	0.39	1.06 ± 0.15	1.98	0.42	1.05 ± 0.14	1.72	0.43	1.14 ± 0.17	1.93	0.23	
Compact. 15 cm (MPa)	6.35	0.006	1.18 ± 0.15b	1.94	0.52	1.23 ± 0.17b	2.16	0.53	1.43 ± 0.18a	2.44	0.34	
Soil moisture (%w/w)	2.77	0.081	26.7 ± 1.5	38.2	15.1	26.8 ± 1.7	45	16.3	24.1 ± 1.4	34.7	14.7	

Plant community dynamics

While their exact components are unknown, the reclamation seed mixes likely included crested wheatgrass and smooth brome (Bromus inermis Leyss.), which were commonly used by the US Forest Service (USFS), Bureau of Land Management (BLM) and other groups during reclamation and revegetation; the USFS used these grasses (despite being exotic and invasive) until recently, while the BLM continues to use crested wheatgrass in severely degraded sites. These grasses are native to Eurasia, and were introduced to grasslands of the US in the early 20th century because of their suitability as cattle forage and their ability to rapidly revegetate abandoned cropland and eroded landscapes. Although suitable as forage during the growing season, these grasses differ from many native species in their resource storage strategies, and consequently provide less nutritious forage (for both livestock and wildlife) throughout the winter, requiring protein supplementation (NRCS, 2006). Despite the likely, though unknown, variation in seeding history, our results demonstrate that plant communities on reclaims were clearly distinct from those on adjacent undeveloped rangeland as we hypothesized (H1). Reclaims had more bare ground, and lower native plant species richness and cover. Given the age of some of these reclaimed sites, it is noteworthy that these plant communities still contain ruderal species (e.g., black medick, curlycup gumweed and dandelion), which are relatively uncommon regionally in undeveloped rangeland communities (Espeland & Perkins, 2017). After an initial increase, ruderal species are expected to decline as the restoration environment becomes more competitive (e.g., Pywell et al., 2003; Espeland & Perkins, 2017); relatively higher ruderal cover in reclaimed sites suggest that competitive dynamics do not drive the low cover and richness of native species.

Figure 4 Boxplots comparing soil abiotic factors between reclaims (“PAD”) and adjacent, intact rangeland 50 m and 150 m from reclaim edges.

Reclaims had reduced native plant species cover, especially for threadleaf sedge and blue grama, the two most common species on undeveloped rangeland. Native plant establishment onto reclaimed sites may be impaired due to a combination of dispersal and seed limitation as well as a reduced capacity for establishment. Because blue grama establishes easily when seeded (Espeland et al., 2017), it is unlikely that it is limited by microsite availability. For this and similar species, their absence from reclaims may simply result from dispersal limitation and suggests these plants may recover their abundance on reclaims only when seeded. We also found inland saltgrass (Distichlis spicata [L.] Greene) on reclaims—it is often found in reclamations but is rarely planted due to high dormancy and seed expense. The presence of this species when absent from surrounding rangeland would suggest effective long distance dispersal to suitable habitats. Native species in the Great Plains have different dispersal abilities that underlie the degree to which propagule supply drives their colonization of reclamations, and soil factors may influence the suitability of habitat conditions and establishment success. Similar to our results on reclaimed well sites, roadside reclamation in this region often fails to recover the full suite of native plant species found in surrounding rangeland (Simmers & Galatowitsch, 2010), but whether this lack of recovery is due to dispersal or establishment limitation is unclear.

Table 3 ANOVA and regression results for nematode community abundance.

All ANOVA testing conducted on 2 and 26 degrees of freedom with sampling location (reclaim, 50 and 150 m native prairie transects) as a fixed effect. Significant ANOVA results indicate response variables differ across sampling locations. For regression models investigating interactions between nematodes, plant and soil factors, CAFI, AGCR and BOGR correspond to species codes for the plants threadleaf sedge, crested wheatgrass and blue grama, respectively.

Response variable	F-statistic	P	Initial regression model	Final model	P	R2	
Total nematodes	3.4	0.049	CAFI + Salts + pH + Bare	pH − Bare	0.008	0.22	
Bacterivores	0.78	0.47	AGCR + SOM + Litter + Salts + pH	SOM	0.01	0.15	
Fungivores	1.96	0.16	SOM + Salts + pH + Litter	SOM	0.006	0.18	
Root herbivores	2.42	0.11	Salts + pH + Bare	pH − Bare	0.006	0.19	
Omnicarnivores	4.98	0.015	BOGR + CAFI + Salts + pH	CAFI	0.004	0.19	

Figure 5 Boxplots comparing select nematode groups between reclaims (“PAD”) and adjacent, intact rangeland 50 m and 150 m from reclaim edges.

Note that as omni-carnivores are comparatively less common members of the nematode community relative to other groups (such as root herbivores) their abundances are an order of magnitude lower than those of total nematode abundances.

Figure 6 Final fitted model used to estimate strengths of interactions between plant, soil and nematode factors.

Solid lines indicate significant (P < 0.05) interactions and dashed line indicates nonsignificant interaction, with values for estimates of standardized path coefficients.

Native species generally do not move onto reclaims, and undesirable plants do not move from reclaims into adjacent rangelands, contrary to our second hypothesis that undesirable plants would increase on intact rangelands with time. Given the amount of time available for these plants to have dispersed from older reclaims into adjacent rangeland communities, it is likely that intact rangeland plant communities in our study are sufficiently competitive (or microsite availability is sufficiently limited) to have prevented establishment of undesirables such as crested wheatgrass, confining them to the reclaims. The only effect of time since reclamation was an interaction between distance from reclaim edges and time, demonstrated by a decrease in exotic plant richness on the 150 m transects with time. Competition may limit the establishment and spread of undesirable species from reclaims into undeveloped rangeland, as these transects are furthest from direct impacts of energy development. A caveat to this is that invasive plant cover did not differ between sampling locations in our study, primarily due to the prevalence of Kentucky bluegrass in rangelands of western North Dakota. This invasive grass was relatively common on our undeveloped rangeland locations but almost entirely absent from reclaims, suggesting that environmental filters may further limit plant establishment in reclaims.

Soil characteristics

Soils on reclaims differed from those of undeveloped rangeland, which supports our third hypothesis. While reclaim soils had greater CEC (generally used as a measure of soil fertility), they also had higher concentrations of salts (sodium and calcium); this increase in base cations also resulted in an associated increase in pH. Soil management may be one cause of the greatly increased salt content of reclaim soils. During well construction, topsoil is scraped from the site and stockpiled nearby. The depth of topsoil excavation is determined to avoid the subsurface concentrated salt layer. However, subsoils are also stockpiled for berm construction and contouring the final reclamation; hence, subsoils are mixed as well as topsoils. The concentrated salt layer present in either of the soil layers is distributed throughout the soil volume during reclamation. Because the layer is no longer concentrated, plants cannot avoid it via root growth. Salinity also influences pH, one of the strongest environmental factors governing soil microbial community composition and structure (Fierer & Jackson, 2006). Given the levels of salinity observed on reclaims, it is possible plant recovery is limited both directly via the influence of salinity and indirectly due to altered microbial contributions to nutrient cycling dynamics (Emam, Espeland & Rinella, 2014). There does not appear to be any change in soil conditions with time since reclamation except for an increase in soil compaction at 7.5 cm. This increased compaction occurred in all transect types and likely reflects cattle grazing or some other factor common to these landscapes. Soil texture on reclaims does not vary with time; the lower proportion of silt in reclaim soils relative to undeveloped rangelands suggests that losses to erosion may occur early in reclamation.

Nematode community dynamics

We found few differences in nematode communities between reclaims and adjacent rangeland. Total nematode abundances were lower on reclaims, as were abundances of omnicarnivores. Part of this variation is likely due to greater prevalence of bare ground on reclaims relative to surrounding rangeland (Wall-Freckman & Huang, 1998). This limits feeding substrates for root herbivorous nematodes and root exudates that support the bacterial and fungal populations upon which bacterivorous and fungivorous nematodes feed. Despite increases in omnicarnivorous and total nematode abundances from reclaims to 150 m transects, it is unclear whether these changes are due to movement of nematodes onto reclaims as only bacterivorous nematodes changed with time since reclamation. This increase with time was found across all transect types. While these differences partly support our fourth hypothesis (H4) that nematode community recovery would be linked to soil conditions and plant community composition, NMS ordination and PERMANOVA results indicate that the trophic structuring and composition of nematode communities are not significantly dissimilar between reclaims and undeveloped rangelands. These results suggest that, contrary to our hypothesis (H5) that nematode communities would recover more slowly than plant communities, nematode community functional composition may be largely recovered following reclamation, while soils, plant community composition, and bare ground show no signs of recovery to reference conditions.

Soil communities along a mining chronosequence in Wyoming also appear to recover rapidly, as communities in even young sites (2 –5 years post-mining) were found to be similar to reference sites (Frouz et al., 2013). The authors ascribed these results to revegetation rapidly recovering climax plant communities, dominance of the nematode community by root herbivores and the broadly similar soil conditions across the chronosequence (Frouz et al., 2013). In our study, however, soil conditions and the plant community are distinct between reclaims and surrounding rangeland, even after comparatively greater periods of time. It may be that in northern plains rangelands the presence of plants alone is sufficient for nematode colonization and persistence regardless of the dissimilarity between the two plant communities. This suggests that within more northern prairies, plant species identity may not matter as much as the presence of the rhizosphere and its associated resources; the observed impacts of bare ground but not plant cover type on the nematode community support this possibility. Similarly, a study in New Zealand grasslands found only weak and inconsistent effects on nematode communities in response to selective removal of C3, C4 or all plants, with the strongest observed responses found in bare ground devoid of plant cover (Wardle et al., 1999).

It is important to note that our study did not examine changes in nematode community composition at the genus or species level, so it is possible that while functionally similar, the nematode communities found on reclaims may be taxonomically distinct from those on adjacent rangeland. Nematode functional-level data are often used in ecological studies because they reflect ecological processes and are considerably less time-intensive to collect than species-level data. However, functional groups may not always capture shifts in community dynamics in response to environmental gradients. For example, a study by Porazinska et al. (2003) found that while nematode trophic groups were resistant to changes in plant diversity, abundances of one root herbivorous nematode genus did differ across treatments. Future work may therefore benefit from finer taxonomic detail, either at the family level and employing various index approaches (such as the Maturity Index; Bongers & Ferris, 1999) or through the use of new DNA sequencing methods (such as in Ramirez et al., 2014). Despite our lack of finer taxonomic detail, the apparent recovery of nematode community functional composition in our system is promising, especially given the importance of functions nematodes provide to terrestrial ecosystems (Lavelle et al., 1997; Sylvain & Wall, 2011).

Soil-plant-nematode interactions

Regressions and SEM illustrate how soil conditions and the biotic community (both plants and nematodes) interact. These analyses highlight the importance of soil conditions to reclamation success, as soils were the most important factor found to explain observed responses in the biotic community in a majority of cases. Regression reflected ANOVA results and revealed salts and SOM were related to the cover of most plant groups. High sodium concentrations in reclaim soils suggest that environmental filters limit the ability of native plant species to colonize and/or establish in the reclaimed area from adjacent, intact landscapes. Increased soil salinity has been well documented to limit plant growth through inhibited nutrient uptake (Hu & Schmidhalter, 2005) and restoration of extremely saline soils may be technically difficult or costly (see Barrett-Lennard, 2002; Pannell & Ewing, 2006). The influence of SOM on plant communities was also found using SEM. Other work in this region has highlighted the importance of SOM in slowing the recovery of reclaimed sites to reference conditions: Viall et al. (2014) observed reduced SOM content on reclaimed road soils, and they proposed that this limited soil microbial community development and decreased the habitat suitability of reclaims for plant production. While we did not observe differences in SOM between reclaims and undeveloped rangeland, it is possible that SOM levels interact with salt-induced nutrient limitation to influence plant community dynamics. The strong influence of soil characteristics on native plant establishment in reclaims may be limited by environmental filters as well as dispersal (discussed above). Restoring plant communities to reference conditions on these sites will require multiple approaches to be successful. This may entail several seed applications with increased species diversity (although repeated seeding with similar seed mixes did not appear to influence richness in a greenhouse study; Wilsey & Stirling, 2007) and mitigating salinity either by excavating the salt layer or by phytoremediation (e.g., Aschenbach, 2006). In contrast to plant communities, nematode communities responded less strongly to soil conditions and appear to be affected more directly by shifts in resource availability. Regression showed nematode abundances were weakly related to SOM levels, but SEM showed a significant effect only of bare ground on the nematode community. SOM and bare ground were highly correlated (Pearson r =  − 0.43, P = 0.005). It is likely that these results indicate the importance of SOM and bare ground to the detrital energy pathway (which includes bacterivorous and fungivorous nematodes) and root resources critical to the herbivory energy pathway components of the soil food web (e.g., Moore & De Ruiter, 2012). Although root herbivorous, fungivorous and bacterivorous nematode abundances did not differ significantly between reclaims and undeveloped rangeland, they did trend lower on reclaims. Across these three groups, the trend in reduced abundances may lead to reduced numbers of omnicarnivores due to prey limitation and consequently to reduced abundances of the overall nematode community present on reclaims.

Conclusion

Hilderbrand, Watts & Randle (2005) note that attempting to create a carbon copy of reference ecosystems through restoration is likely an unachievable goal because the conditions present at the start of reclamation (landscape degradation) are fundamentally different from those governing community succession after less dramatic disturbances. The authors instead suggest that reclamation should aim to recover the ecosystem to the best possible extent, opting for functional rather than taxonomic recovery when necessary, with resilience to future disturbance in mind (Hilderbrand, Watts & Randle, 2005). This would enable reclaimed areas to provide a comparable suite of functions and minimize impacts of fragmentation and invasive species (see Allred et al., 2015) while dynamics of plant community assembly and soil formation proceed over time to restore species diversity and cover through natural processes. Our multi-trophic approach demonstrates that soils, plants, and nematodes recover to reference states at different rates. Soils were linked to the slow recovery of the plant community. Plant presence rather than plant identity mattered more to the nematode community, and soil characteristics did not appear to limit nematodes. Our results suggest increased seeding rates and diversity of native plant species will improve recovery. Additionally, the incorporation of soil amendments to address salinity and nutrient availability may help mitigate environmental filters that contribute to poor native plant establishment on reclaims and inhibit successful recovery of ecological communities and their function. Our study shows that every community may have an idiosyncratic response to reclamation, contrary to the “field of dreams” presumption (Hilderbrand, Watts & Randle, 2005) prevalent in both reclamation and restoration practice.

Supplemental Information

Table S1 Most commonly sampled plants, with authorities and groupings used in analyses

Designations include whether a species is native (N) or exotic (E), with invasive (I) or ruderal (R) status indicated in parentheses where appropriate. Functional groups include cactus (C), forb (F), grass (G), legume (L), sedge (Se), shrub (S) and sub-shrub (Sub). Asterisks denote an indicator species determined from indicator species analysis.

Click here for additional data file.

Table S2 Plant indicator species by sampling location

Exotic species in bold and functional groups denoted by F (forb), G (grass) or S (shrub). Ruderal plant species are denoted by ‡ and invasive species are denoted by *.

Click here for additional data file.

Table S3 Mean (± SE) values for nematode trophic groups by sampling location

Bold text denotes significant differences between sampling location (letters denote post-hoc test groupings; see text).

Click here for additional data file.

Table S4 Coefficient estimates and standard errors for direct interactions between plant, soil and nematode factors in final SEM

Click here for additional data file.

Figure S1 Histogram showing frequency of transects by time since sites released from bond

Click here for additional data file.

Figure S2 Proposed conceptual models hypothesizing a nearly saturated interaction web relating soil factors to plant and nematode community structure, and interactions between the plant community and nematode community

The conceptual model is constructed on hypotheses related to environmental filtering (impacts of soluble salts) and resource availability (SOM, litter and plant factors for nematodes).

Click here for additional data file.

We thank M O’Mara for her assistance with site selection and sampling, N Davidson, K Hauri and C Schilling for assistance with field sampling, the Watford City office of the U.S. Forest Service for their assistance with site selection and background information and J Gaskin for helpful comments on an earlier manuscript draft.

Additional Information and Declarations

Competing Interests

Author Contributions

Data Availability

The authors declare there are no competing interests.

Zachary A. Sylvain conceived and designed the experiments, performed the experiments, analyzed the data, contributed reagents/materials/analysis tools, prepared figures and/or tables, authored or reviewed drafts of the paper, approved the final draft.

David H. Branson and Tatyana A. Rand conceived and designed the experiments, performed the experiments, contributed reagents/materials/analysis tools, authored or reviewed drafts of the paper, approved the final draft.

Natalie M. West and Erin K. Espeland conceived and designed the experiments, performed the experiments, contributed reagents/materials/analysis tools, prepared figures and/or tables, authored or reviewed drafts of the paper, approved the final draft.

The following information was supplied regarding data availability:

Sylvain, Zachary A; Branson, David H; Rand, Tatyana A; West, Natalie M; Espeland, Erin K. (2019). Data from: Decoupled recovery of ecological communities after reclamation. Ag Data Commons. http://dx.doi.org/10.15482/USDA.ADC/1503836.

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
