# Peer review of "Decoupled recovery of ecological communities after reclamation"

_PeerJ, doi:10.7717/peerj.7038_

## Round 0.1 · original submission · Major Revisions

After reading through the reviews and the manuscript myself, I think this manuscript has the potential to make an important contribution to the fields of restoration and conservation after some careful clarification and revision to address the comments made by the reviewers.

Reviewer 1 ·

Basic reporting

The manuscript reports on plant, soil, and nematode responses to reclaimed rangelands. The manuscript is well written and follows appropriate formatting guidelines. The manuscript has sufficient background and represents a self-contained unit of publication. This manuscript should be of interest to the broad readership of PeerJ.

Experimental design

The design seems appropriate. However, there are a few points that need clarification.

Lines 205-206: Please also state if type III sum of squares was used (this is the default). This seems most appropriate for your design.

Line 207: By “sampling location” do you mean reclaimed/remnant prairie? Using the terms location and site in the same model description could be confusing.

Lines 207: After you performed the mixed-models analysis, was the model homoscedastic? Or were there changes in the spread of residuals? You should confirm the Pearson residuals are not changing in their spread using plot(model). This is a more important assumption than normality.

Line 212: This is confusing. I think you meant to say post-hoc multiple comparisons using the Holm method were performed. The Tukey test is another multiple comparison method.

Validity of the findings

The findings in this manuscript seem valid.

Additional comments

Line 25: Should there be keywords after the abstract?

Line 54-55: Might the slow recovery be due to dispersal limitations? How might the soil microstructure and plant community influence dispersal?

Line 167: I think Table S1 should be part of the main manuscript, not a supplemental table. It could be combined with Table 2. Also a measure of soil moisture (such as gravimetric water content) should be reported with soil compaction test.

Line 215: Why not put the continuous predictors in the same model as the categorical predictor?

Line 229: Make the NMDS and PERMANOVA a new paragraph. Consider performing a PERMDISP as well. It seems the two groups in Fig. 3 are overlapping (perhaps a spurious PERMANOVA result because of differences in dispersion of the two groups).

Fig. 3: Add a legend for the two groups.

Line 356: Remove the comma between adjacent and intact

Line 435: Reorder the sentence so that the author is not the subject of the sentence.

Lines 443-446: This is interesting that the plant presence rather than plant identity could be important for nematodes in your study areas. Perhaps you could expand a bit more. This could be because there is little variation in plant species present. If there is considerable variation in plant species, then perhaps many nematode groups can respond to a wide variety compounds deposited in the rhizosphere.

Lines 482-483: Below is a reference that might be useful here that supports repeated seeding can increase plant richness

Wilsey B, Stirling G (2007) Species richness and evenness respond in a different manner to propagule density in developing prairie microcosm communities. Plant Ecol 190:259–273

·

Basic reporting

Writing is clear and English correct. Literature citations are appropriate and sufficient. Article structure is good, figures and tables professionally done.

There are some changes that could clarify the captions and interpretation of the Figures and Tables:
Tables 1-3: The caption could include a reminder that sampling location is the main fixed effect and clarify what a significant response means in this analysis.

Fig. 3: I might suggest a legend rather than just explaining in the caption, but totally up to the authors.

Fig. 5: I would suggest pointing out the difference in scale between the panels here. I see why it is the way it is, but it’s also worth pointing out that the omnicarnivores are an order of magnitude lower than total abundance (gives a sense of their contribution to the overall community).

The Raw data are currently included as supplemental material. I would encourage the authors to consider a public database like Dryad to share the data. I also could find no link to the R code for the statistical analysis, although the is mentioned in the supplemental material.

Results all addressed specified hypotheses and were synthesized into an informative SEM model. The synthesized SEM also provides a compelling case for including 5 hypotheses in the manuscript, which could be overwhelming without a concrete synthesis.

Experimental design

This is an original data manuscript and fits well within the Aims & Scope of the journal.

The research questions are well-defined and addressed appropriately. This work does address a gap in scientific knowledge, as very little work in restoration ecology has synthesized above- and belowground communities across multiple trophic levels. The work on reclaimed well heads in the Brakken oil field is also an important contribution to address the future of well-heads as they are retired in this region. The importance of considering soil invertebrates like Nematodes, which uniquely span multiple trophic levels could be articulated even more in the manuscript.

The investigation was rigorous, using respected approaches to sampling and analysis. I found no ethical concerns.

Some additional information could be included to clarify the experimental design in the Methods.
i. I strongly recommend a table, even if supplemental the showed site, age class, and more clearly represented what was considered “undisturbed prairie” (e.g. the transect 100m from the reclaim, both the 50 and 100?) Not all readers will realize the reclaimed well heads are surrounded by undisturbed prairie, and that the transects moving away from the reclaimed area are representing undisturbed prairie. This would help tremendously with clarifying the fixed effects categories and degrees of freedom reported with the ANOVA statistics. For example, I was unsure where the degree of freedom of 26, reported in Tables 1-3 is coming from. I understand the 2 (reclaim/50m/150m distinction), but total N=42. Were only 2 of the 3 transects included in the ANOVA (28-2=26)? Please clarify if I’m missing something.
ii. Building from (i), it is challenging to understand how the sites are distributed across the 33 year consequence. This adds confusion when the location*time interaction is discussed (lines 98-100, 2017-220, 264-269, 387-389, 440-441). Consider how including a table of the sites or changes to the text might clarify these sections.
iii. Site was used as a random factor, and that is fine. Was there any consideration of using time as a random factor, to account for potential changes over time in the ANOVA model?

Validity of the findings

As mentioned above, this work addresses a clear knowledge gap and the findings provide important context for not only understanding how soil and plant communities recovery in reclaimed oil and gas drill sites. The paper builds the ecological knowledge gap first, and it might be helpful to identify what is meant by "well" earlier in the text, even in the abstract, as not all readers will jump to the understanding that this means natural gas or oil well.

The data are robust and controlled. The data are statistically sounds, although some clarification in description and reporting of the statistics would improve the manuscript.

The conclusion evaluates the hypotheses and stays well within the frame of inference from the data.

I have no concerns about speculation arising from the data.

Additional comments

Overall, very nice paper. As mentioned above, there are some changes that could be made to clarify the experimental design, particularly the time aspect. In addition, clarifying that wells and reclaims refer to gas and oil wells might be helpful in the abstract and early in the Introduction, just to help out readers, particularly international readers, unfamiliar with what is meant by energy development in the Great Plains.

·

Basic reporting

no comment

Experimental design

no comment

Validity of the findings

no comment

Additional comments

This study examines plant and soil community response to disturbance associated with oil extraction in the Bakken oil fields of western North Dakota. Adding the soil community analysis is the novel element in this project. Therefore I focused on Hypothesis 4 and 5. Hypothesis 4 predicts that the nematode community recovery on reclaimed sites would be linked to soil factors and plant community. This is a reasonable prediction since there is an increasing amount of information that microbial feeding nematodes respond positively to plant biomass. Plant feeders have been shown to respond positively to plant diversity and presumably resource quality reflected by C/N ratios. (Cortois et al, 2017; Cesarz et al, 2017). Hypothesis 5 predicts that the nematode community would lag behind that of plant communities due to slower dispersal. This prediction, I believe is probably contingent on geographic scale. Certainly, plant feeding specialists are dependent on the presence of their hosts. But some species of plant-feeding grassland nematodes have broader host-ranges, feeding on a number of native and possibly introduced grass species. A number of the plant-feeding genera, e.g. Paratylenchus, Helicotylenchus, ,and Quinisulcius all members of the mixed-grass prairie community, appear to go in and out of an anhydrobiotic state rapidly, and probably are dispersed by wind-blown soil in dry conditions or in surface water during heavy rains. Many microbivores have phoretic associations with insects. Nonetheless, both hypotheses are testable. Basically the results did not support a specific linkage nor a time lag in the relationship between nematode communities and soil conditions or plant communities. While the authors can speculate about possible fewer qualitative requirements in northern prairie nematode communities, I think they are closer to the answer in the paragraph starting on line 450. The trophic group analysis is a very coarse measure of nematode community structure. Since the analyses were probably made at the family or superfamily taxonomic level, the reclaimed and rangeland soil nematode communities could differ considerably in taxonomic composition. The limitations of the trophic group analysis, which has been a standard form of ecological analysis for years, is a compromise due to the time-consuming taxonomic analysis necessary to get a more refined understanding of nematode community structure. The authors point out this trade-off. However, one concern I have that is not addressed by the authors, is the method of nematode extraction. Many ecologists persist in conducting Baermann funnel nematode extractions when it is well known that funnel extraction bias nematode recovery in favor of fast moving, active nematodes. Several prominent groups of plant feeding nematodes known to be abundant in grasslands will be underrepresented in funnel extractions. Obviously it will bias reference and disturbed sites equally, but the resulting trophic group comparisons with other studies is skewed. Overall I think the study with regards to the plant community and soil conditions is probably solid, the verdict on the nematode communities is still an open question. Greater taxonomic resolution, either by higher resolution microscopic analysis or molecular approaches are probably necessary to resolve the issue.

---

## Round 0.2 · Minor Revisions

Thank you for your careful revisions in response to previous reviews. At this point there are only some very minor issues (particularly with figure legends) that should be addressed, as per Reviewer 2.

Reviewer 1 ·

Basic reporting

No comment.

Experimental design

No comment.

Validity of the findings

No comment.

Additional comments

The authors have addressed all my concerns about the manuscript.

I have one minor editorial note: on line 142 the sites can be referred to as reclaims because the abbreviation was indicated earlier in the methods.

·

Basic reporting

Lingering revisions:
• Table captions: The authors state that they made changes to the text to specify the fixed effect in Tables 1-3 and clarify what the response meant, but it seems the caption text is identical to the original manuscript.
• Figure 5: Authors state that a change was made to the legend to highlight the scale difference in panel A and B; however, I do not see a legend at all on this figure.

Experimental design

No further clarifying explanation required.

Validity of the findings

No further revisions necessary.

Additional comments

Dear Sylvan et al.,
I have reviewed the revision of “Decoupled recovery of ecological communities after reclamation” submitted to PeerJ (peerj-30920-v1). Overall, I find that you have made careful and thoughtful revisions. I noted two places where your response indicated changes had been made, but I failed to find them in the revised manuscript:
• Table captions: The authors state that they made changes to the text to specify the fixed effect in Tables 1-3 and clarify what the response meant, but it seems the caption text is identical to the original manuscript.
• Figure 5: Authors state that a change was made to the legend to highlight the scale difference in panel A and B; however, I do not see a legend at all on this figure.

Perhaps the wrong files were incorrectly uploaded during the revision submission process? It seems the concerns were addressed, I just could not find the changes. Thank you for your time and effort on this manuscript.

---

## Round 0.3 · accepted · Accept

We appreciate the work you've put into revising the manuscript and think it will make a good addition to PeerJ.